

# CuentosIE: can a chatbot about "tales with a message" help to teach emotional intelligence?

Antonio Ferrández[1], Rocío Lavigne-Cerván[2], Jesús Peral[1], Ignasi Navarro-Soria[3], Ángel Lloret[1], David Gil[1] and Carmen Rocamora[4]

[1] Department of Languages and Computing Systems, Universidad de Alicante, Alicante, Spain
[2] Department of Developmental and Educational Psychology, Malaga University, Malaga, Spain
[3] Development Psychology and Teaching Department, Universidad de Alicante, Alicante, Spain
[4] Nursing Department, Universidad de Alicante, Alicante, Spain

## ABSTRACT

In this article, we present CuentosIE (TalesEI: chatbot of tales with a message to develop Emotional Intelligence), an educational chatbot on emotions that also provides teachers and psychologists with a tool to monitor their students/patients through indicators and data compiled by CuentosIE. The use of "tales with a message" is justified by their simplicity and easy understanding, thanks to their moral or associated metaphors. The main contributions of CuentosIE are the selection, collection, and classification of a set of highly specialized tales, as well as the provision of tools (searching, reading comprehension, chatting, recommending, and classifying) that are useful for both educating users about emotions and monitoring their emotional development. The preliminary evaluation of the tool has obtained encouraging results, which provides an affirmative answer to the question posed in the title of the article.

## INTRODUCTION

The first approach to the definition of emotion emerged a century ago, proposed by *James (1884)* and *James (1890)*. Since then, various authors (*e.g.*, *Goleman, 1995*; *Mayer & Salovey, 1990*; *Morgado, 2007*; *Cantón, Cortés & Cantón-Cortés, 2011*) have concurred with James that emotion is an innate mechanism that allows individuals to adapt to their environment. Specifically, the popularization of the term "emotional intelligence" is attributed to the work of *Goleman (1995)*, which analyzes the ability to motivate oneself, persevere in the face of frustration, delay gratification, control one's impulses, regulate one's moods, prevent distress from interfering with one's rational faculties, and empathize with and trust others.

Neuroscience has enabled the direct and in-depth study of brain areas involved in emotional processing, both involuntary and controlled. These emotions arise in the first months of life and gradually change due to the maturation of other cognitive processes, as well as learning (*Cantón, Cortés & Cantón-Cortés, 2011*) in social interaction situations,

Corresponding author
Antonio Ferrández,
antonio@dlsi.ua.es

which facilitates the development of self-regulation strategies that enable individuals to adapt to their environment (*Thompson, Lewis & Calkins, 2008*).

Throughout development, children are exposed to peer relationship situations that require them to balance their emotional and cognitive processes in order to respond effectively to the demands of their environment. The development of social cognition, which is the ability to identify and understand social situations (*Uekermann et al., 2010*), requires the coordinated functioning of cognitive and affective elements (*Roselló et al., 2016*). The coordinated work of these elements allows individuals to acquire and improve more complex skills (social skills), which promote the generation of specific responses in situations of interaction with peers.

However, emotional education remains a pending issue in our society, despite its potential benefits in addressing many current problems, such as bullying, suicide, gender violence, stress, anxiety, depression, anorexia, discrimination, and autism.

Chatbots are programs that simulate having a conversation with a person, and their use has become widespread in today's digital society. Their expansion and implementation as a communication channel is justified by their intrinsic characteristics of wide availability and anonymity in their interaction through the web. People enjoy talking to chatbots because they do not judge and are always patient, even when people talk for a long time or repeat themselves (*Fryer & Carpenter, 2006*; *Hill, Ford & Farreras, 2015*). Well-known virtual assistants (*e.g.*, Siri, Alexa, *etc.*) are widely used in different domains, such as education and customer service.

In this article, we present the web chatbot CuentosIE (TalesEI: chatbot of tales with a message to develop Emotional Intelligence), which has educational purposes on emotions. Addressing this aim through "tales with a message" is justified by following the millenary tradition of the human being, which has been shown to be highly effective in transmitting and understanding knowledge in a way that is easily understood thanks to its simplicity through its moral or associated metaphors.

Regarding the target and beneficiary groups of CuentosIE, both students, teachers, patients, and mental health professionals would be included, since one of the main difficulties encountered by psychologists in their consultations is getting patients to truly express what they feel (and sometimes think) and to get to the root of their problems (*Chan et al., 2016*; *Wallin, Mattsson & Olsson, 2016*). Thanks to the anonymous nature of the internet, this chatbot is intended to help any type of user.

This article presents the following contributions:

(C1) The design and development of a chatbot for the teaching of emotions (CuentosIE), which uses tales with a message as its knowledge core. CuentosIE was initially developed for Spanish tales, but it can be ported to other languages since its components are widely available.

(C2) The proposal of an architecture that overcomes the hallucination issue in modern chatbots such as ChatGPT and BARD, which could be critical in healthcare, particularly in mental health. For example, *Eshghie & Eshghie (2023)* use ChatGPT as a therapist assistant "without providing explicit medical advice"; they highlight ChatGPT's limitations in terms of "recalling conversations from previous sessions" and its

inability to "read non-verbal cues such as body language or facial expressions". The latter limitation is also corroborated by *Carlbring et al. (2023)*, who conclude that "a conversational agent mimicking empathy and responding appropriately may not be enough".

(C3)  This architecture coordinates the interaction of the chatbot with an information retrieval system (to allow the user to search for tales), a reading comprehension question generator (to help the user understand the tales), and an emotion classifier system (to detect the user's emotions based on how they interact with CuentosIE, in order to recommend tales related to these emotions). Transformer technology is used in a controlled way in the interaction with the user in the reading comprehension question generator and the emotion classifier.

(C4)  The selection, classification, and labeling of tales according to emotions and psychological themes carried out by our psychologist authors, all of which were selected from the wide variety of websites dedicated to the publication of tales, as well as scientific works that support the usefulness of these tales (*Färber & Färber, 2015*; *Odabasi, Karakus & Murat, 2012*; *Kulikovskaya & Andrienko, 2016*). In this way, we avoid the hallucination problems of ChatGPT and BARD, since the conversations will be restricted to these pieces of knowledge.

(C5)  We also overcome the previously mentioned issue about "recalling conversations" since all the user's interactions with the chatbot are stored as XML files, which can be used to detect sensitive situations of depression (*Havigerová et al., 2019*), suicide (*Boggs & Kafka, 2022*), or bullying (*Bayari & Bensefia, 2021*).

(C6)  The evaluation of the tool has obtained encouraging results.

The remainder of this article is organized as follows: 'Background' contains an overview of related literature. In 'Architecture of CuentosIE', our proposal is fully described. 'Experimentation' presents the experimentation carried out. The main conclusions and future lines of research are drawn in 'Conclusions'.

## BACKGROUND

In this section, we focus on the study of emotions, particularly through tales. Subsequently, we analyze previous work on chatbots that address this topic.

### Background on dealing with emotions

Emotions are an innate mechanism triggered by environmental stimuli, which generate a specific physiological response in the form of involuntary and automatic processes (*Damasio, 1996*). However, it is possible for such processes to be triggered after our brain has carried out a conscious and voluntary evaluation of the situation in which we are immersed. This is not the case for all emotional reactions or behaviors experienced by humans.

According to this way of understanding emotions, we can differentiate between primary and secondary emotions. Primary (innate) emotions: These emotions are present from birth or the first months of life due to their biological conditioning (*Cortés, Cantón & Cantón-Cortés, 2011*). The mechanism that activates these emotions is basic: environmental

signals or stimuli are detected by the sensory cortex and processed by the limbic system (specifically by the amygdala), which is responsible for activating a physiological state and altering cognitive processing according to the emotion that matches the information generated by the external stimulus. The goal of these emotional responses is often linked to survival. The reactions experienced by the body can be associated with the specific object that caused them, creating a propositional representation of the relationship between the emotional state and the stimulus that triggered it. This would help us to predict the presence of that stimulus in a given context and even anticipate our response in future scenarios (*Damasio, 1996*). Primary emotions include happiness, anger, sadness, and fear.

Secondary (non-automatic) emotions: These emotions are initiated in the same brain mechanisms as primary emotions, but they also require thought processes that occur in parallel to them. According to *Damasio (1996)*, the person already has a series of mental images of stimuli related to emotional reactions, which have been organized by thought, after carrying out a cognitive evaluation of previous situations or experiences. These are emotions like shame, blame, pride, and hatred.

Research on natural language processing (NLP) techniques in emotion processing mainly deals with the detection and classification of emotions in multimodal (*e.g.*, *Pepino et al., 2020*) and written texts (*e.g.*, *Mihalcea & Liu, 2006*; *Li & Xu, 2014*) using machine learning approaches (*e.g.*, support vector machine or k-nearest neighbor). The classifiers vary across the emotion class taxonomy, from simple taxonomies as the ones used in sentiment analysis applications (*e.g.*, *Chaumartin, 2007*), to fine-grained emotion classification (*e.g.*, the 10 emotions in *Tokuhisa, Inui & Matsumoto, 2008*; or the 14 emotions in *Santos, Ong & Resurrección, 2020*). The machine learning processing and evaluation is run on a variety of datasets tagged with emotions, such as the Emognition dataset specialized in emotion recognition with self-reports, facial expressions, and physiology using wearables (*Saganowski et al., 2022*); or the K-EmoCon, a multimodal sensor dataset for continuous emotion recognition in naturalistic conversations (*Park et al., 2020*).

The use of tales with morals as a tool for working with emotions has proven to be effective for teaching social, affective, emotional, and moral aspects, allowing users to establish cause-and-effect links and develop resilient behaviors, among other things (*Färber & Färber, 2015*; *Kulikovskaya & Andrienko, 2016*; *Odabasi, Karakus & Murat, 2012*). If we digitize and dynamize this process, giving the user a leading role by allowing them to interact with the tool and select the most appropriate tale for their current emotional state, we believe that the value of the activity will increase substantially. Furthermore, we can design this tool with universal learning design in mind, *i.e.,* guaranteeing participation and accessibility to all individuals. Similarly to *Sánchez Calleja, Benítez Gavira & Aguilar Gavira (2018)*, we consider that using this type of resource allows us to show tales to the user through texts that can be translated into different languages, and can be shown both visually and audibly, with supports such as pictographs and videos.

## Background on chatbots used for emotion processing

The 'Turing Test', introduced by Turing, serves as a pivotal benchmark for determining whether a robot can exhibit human-like behaviour. In this test, an evaluator engages in

a conversation with two interlocutors—an AI bot and a human—through an interface. If the evaluator cannot distinguish which is the bot within a five-minute time interval, the machine is considered to have passed the test. Alan Turing's ground-breaking 1951 theoretical postulate not only tested the intelligent behaviour of machines against humans but also laid the foundation for the development of virtual assistants and chatbots (*Turing, 1951*; *Turing et al., 1952*).

The Loebner Prize competition, initiated in 1991, allows various robots to compete in the quest to pass the renowned Turing Test. For many years, prizes were awarded to the best-performing robots, but as recently as 2014, over two decades after the competition's inception, that a machine successfully passed the test (*Khan & Das, 2017*; *Mauldin, 1994*; *Warwick & Shah, 2014*; *Warwick & Shah, 2016*). This evolution marked a significant milestone in the field of artificial intelligence as the Loebner Prize competition serves as a testament to the persistent pursuit of human-like conversational abilities in machines. This historical backdrop provides essential context for understanding the subsequent development and application of chatbots, particularly those designed for emotion processing in educational settings.

The origins of chatbots are intricately linked to the field of psychology, with the initial strides in their development attributed to *Weizenbaum (1966)*. Weizenbaum created the pioneering program ELIZA, often regarded as the first chatbot in history. ELIZA simulated conversations with a psychologist by identifying keywords in user input and generating relevant questions. Despite its predefined answers, ELIZA conveyed a remarkable sense of understanding to users, serving as a catalyst for subsequent advancements. Building on the foundation laid by ELIZA, Colby introduced PARRY in 1971 (*Colby, Weber & Hilf, 1971*). PARRY adopted a similar architecture but took on the persona of a paranoid patient. Armed with approximately 6,000 patterns for recognizing input elements and a set of open-pattern stock answers, PARRY showcased the diverse applications of early chatbots. Alicebot, created by Wallace in 1995, emerged as another significant milestone. Winning the Loebner award several times (*Wallace, 2009*). Alicebot boasted over 40,000 categories of knowledge, significantly surpassing ELIZA's capabilities. These categories, organized in a tree diagram, facilitated dynamic and engaging dialogues.

Lately, there has been a remarkable development of animated virtual agents—avatars possessing human-like appearance, gestures, and expressions that engage with users. This form of presentation serves to enhance the chatbot's perceived sociability and overall user experience (*Klopfenstein et al., 2017*). These agents have pioneered the way for the design of a diverse array of conversational entities. Among the most widely recognized are those embedded in our everyday devices, including Google Assistant (developed by Google), Siri (developed by Apple), Cortana (developed by Microsoft), and Watson (developed by IBM). These advanced agents seamlessly interact with users through both text and voice, facilitating a range of tasks such as music activation, medical appointment scheduling, answering inquiries, and even ordering food orders for home delivery (*Khan & Das, 2017*). Concurrently, a growing number of companies are integrating chatbots into their websites and social media platforms to provide customers with streamlined access to products and services. For example, in the banking sector, Blue is a customer assistant chatbot in

personal banking for the use of the BBVA app. It is based on predefined questions, and the responses cover both inquiries related to the handling and usage of the application, as well as obtaining personal information about the client's banking status (*Blue, 2024*). Lowe's, the home improvement retailer, has a chatbot on its website that can primarily assist with product-related inquiries, aiming to locate them within the extensive catalog available on the web; it also enables users to find physical stores (*Lowes, 2024*).

While the majority of chatbots have been designed for commercial applications, their utility extends beyond other areas, such as telecommunications, security, tourism promotion or health. DroidPerf, a lightweight Android profiler, develops a communicable bot to perform analysis to uncover memory inefficiencies in Android apps running on the Android Runtime platform (*Li et al., 2023*). EMMA is a virtual assistant developed for the US Citizenship and Immigration Services that assists individuals with requests related to immigration services, green cards, passports, and any services offered by the department (*US Department of Homeland Security, 2024*). Turisme Comunitat Valenciana (Spain) has included a marketing and promotion chatbots to attract tourists and provide information about its destinations, events, and tourist activities (*Santa Pola, 2024*; *Vinarós, 2024*). With respect to health area, a noteworthy non-commercial application lies in their potential as support tools for psychological evaluation and intervention tasks. Regarding mental healthcare, the use of chatbots can be categorized into three primary areas within the psychotherapeutic context: prevention, treatment, and follow-up/relapse prevention of psychological problems and mental disorders (*Bendig et al., 2019*).

In the domain of prevention and detection of mental health-related disorders, two noteworthy works stand out. SentinoBot (*Sentino, 2016*) is specifically designed for psychological evaluation tasks with a focus on assessing personality traits. This virtual agent collects information on key traits—extraversion, responsibility, kindness, neuroticism, and openness to experience—through multiple-choice questions, employing a Likert-type scale for responses. It operates as a virtual evaluator, conducting a structured assessment with closed questions and predefined answers. Another significant contribution is Replika (*Replika, 2021*), a conversational chatbot designed for an initial psychological evaluation (pre-evaluation). Functioning as a virtual friend, Replika engages users in a friendly interaction, prompting discussions about daily activities, hobbies, aspirations, and emotions. Notably, Replika possesses the capability to identify keywords associated with psychological distress. Importantly, it includes a critical feature of referring users to specialized care services when it detects a potentially suicidal attitude. This dual focus on evaluation and intervention showcases the versatility of chatbots in addressing mental health concerns.

In the area of treatment or intervention of mental disorders, we highlight the following works. WOEBOT (*Fitzpatrick, Darcy & Vierhile, 2017*) is a conversational chatbot that performs an interaction like a therapeutic conversation. The authors used the chatbot in a personal development program for students addressing depression and anxiety disorders, grounded in cognitive-behavioral principles. The experiments showed a substantial decrease in depressive and anxiety symptoms in students who tested the chatbot compared to those who used an eBook on depression. SHIM chatbot (*Ly, Ly & Andersson, 2017*),

also rooted in cognitive-behavioral therapy and positive psychology elements, serves as a self-improvement initiative to enhance mental health and reduce perceived stress. The evaluation proved the practicability of the chatbot with a high adherence to intervention completion and noteworthy impacts on mental health and stress levels. GABBY chatbot (*Gardiner et al., 2017*) is a personal development program designed to aid in altering behavior and handling stress independently, drawing inspiration from mindfulness-based stress reduction principles (*Gardiner et al., 2013*). The experimentation demonstrated no significant difference related to perceived stress between the different groups. However, an important conclusion about differences in stress-related alcohol consumption was extracted. The experiments corroborated the practicability of GABBY with respect to adherence, user satisfaction, and the proportion of users from ethnic minorities. MYLO (*Bird et al., 2018*; *Gaffney et al., 2014*) is another chatbot specialized to cope with stress problems; its foundation lies in the principles of the perceptual control theory. It offers a self-improvement program for problem solving related to depression, anxiety, and stress. In the experiments, MILO was compared with ELIZA, with both chatbots leading to the relief of the mentioned disorders. However, MYLO was deemed more beneficial for effective problem-solving.

With regard to other self-improvement programs aimed at promoting of mental well-being, we can mention the following. SABORI (*Suganuma, Sakamoto & Shimoyama, 2018*) is a chatbot for this purpose, Including concepts from cognitive-behavioral therapy and behavioral activation principles. A prospective pilot study examined psychological health, psychological distress, and behavioral activation for all the participants concluding the practicability of SABORI as a prevention program. *Kamita et al. (2019)* proposed a self-mental healthcare chatbot course on the LINE platform, commonly used as a smartphone communication tool. The experiment aimed to compare the chatbot course with a web-based course. The results proved the stress reduction effect and the improvement of user's motivation when using the chatbot course on smartphones.

To conclude the area of treatment/intervention, we mention two additional approaches. PEACH is a chatbot (smartphone application) oriented to personality coaching (*Stieger et al., 2018*). It employs diverse psychotherapeutic mechanisms and micro-interventions, requiring minimal therapist interaction and facilitated by chatbots. It addresses issues such as change motivation, psychoeducation, behavioral activation, self-reflection, and resource activation. A digital coach in the form of a conversational agent will be presented to assist users in reaching their objectives for personality change. The evaluation demonstrated the intervention's effectiveness in the pos *t*-test assessment. Rose (*De Gennaro, Krumhuber & Lucas, 2020*) is a chatbot oriented to treat social ostracism. It provides empathetic answers to help participants who experienced social ostracism recover from the experience. Users predominantly engaged with the chatbot through multiple-choice menus that may be updated depending on the conversation topic. In the evaluation, after the chatbot intervention the participants reported an improvement in their mood.

Finally, chatbots could be used in the future after classical psychotherapy is finished to support follow-up and relapse prevention. They could help to stabilize the effects of

the intervention, facilitating the integration of therapeutic content into daily life, and diminishing the likelihood of regression (*D'Alfonso et al., 2017*).

Regarding chatbots that deal with users' emotions, several examples can be mentioned. Microsoft XiaoIce (*Zhou et al., 2018*) is designed as a social chatbot with the ability to recognize human feelings and states. The emotion companion chatbot EREN (*Santos, Ong & Resurrección, 2020*) uses storytelling to elicit story details from children, by labeling emotions and facilitating a discussion to help the child reflect on these emotions and their related events; and by analyzing the child's narrative.

To conclude this section, we must highlight the emergence of modern chatbots such as ChatGPT and BARD. As mentioned in the introduction section, these chatbots have been used as AI-assisted therapist assistants (*Eshghie & Eshghie, 2023*; *Carlbring et al., 2023*), with the limitations previously enumerated, which we have overcome with the proposed architecture for our chatbot.

# ARCHITECTURE OF CUENTOSIE

In this section, as our contribution (C1) outlined in the introduction, we provide a comprehensive description of the chatbot CuentosIE. Currently, CuentosIE is solely available in Spanish, but it can be readily adapted to other languages due to the availability of the natural language processing (NLP) tools employed in the chatbot. These NLP tools encompass the POS-tagger, parsers, and semantic resources from FreeLing (*Padró & Stanilovsky, 2012*); the Natural Language Toolkit (NLTK, as cited in *Bird, Klein & Loper, 2009*); the spaCy library (*Honnibal & Montani, 2017*); the information retrieval tool (*Ferrández, 2011*); and machine learning classifiers, which are contingent upon the presence of tagged corpora in the target language.

The next five subsections delve into the detailed architecture of the system. The first subsection provides a general overview, while the remaining four subsections expound upon the most crucial modules: the Discourse Manager and Question Generator, which facilitate reading comprehension; the Information Retrieval module dedicated to selecting, classifying, and labeling stories; the Emotion Classifier module; the Emotional Evolution Monitoring module.

## A comprehensive overview of CuentosIE's system architecture

The architecture of CuentosIE is presented in Fig. 1, which depicts its flow of information from (1) to (5), as well as the distinct modules that will be elaborated upon in the ensuing subsections. In flow (1), users access the chatbot through computers and mobile devices at the URL https://oldgplsi.gplsi.es/cuentosIE/index.php, featuring a user interface crafted using HTML, PHP, and JavaScrip. They can register to receive future information about their development in their interactions, as well as to avoid being recommended tales that they have already read or that are not appropriate for their age. The registration process does not require any personal information to maintain the user's anonymity. Users are only asked for their age and gender to collect statistics, data that is housed within a MySQL database. If a user does not register with CuentosIE, they will be identified as a ''non-registered user''. Emotional evolution monitoring (flows (3) and (5) that are explained in

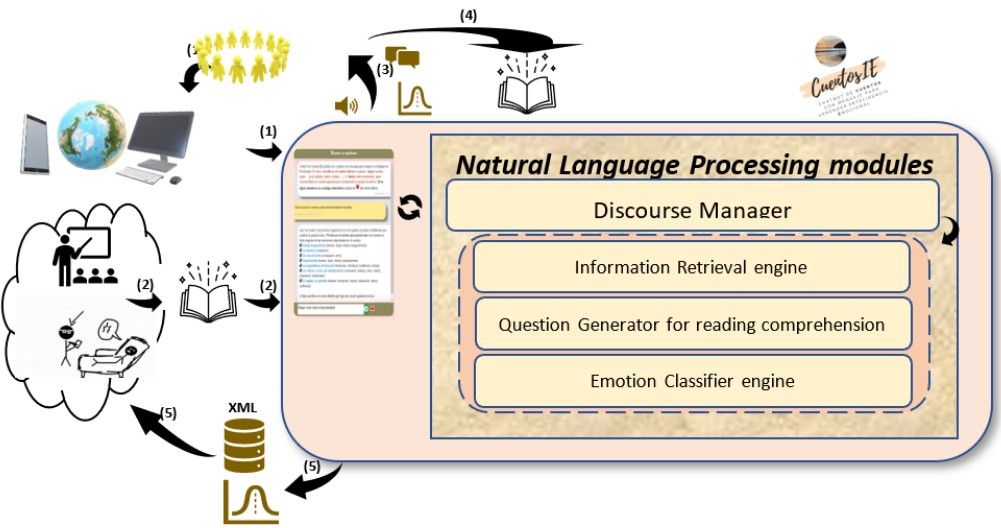

**Figure 1** **Architecture of CuentosIE.**

'The Emotional Evolution Monitoring Module') is only performed on registered users, and their anonymity is preserved because no email address or other identification is required. If a sensitive situation is detected after analyzing the user's interactions, as will be described in more detail in contribution (C5), CuentosIE will activate and display the corresponding alarm signal the next time the user enters the chatbot. In the case of using CuentosIE in a school or by mental health professionals, users can allow teachers or psychologists to know their identification names to monitor their progress.

After the user is identified, the chatbot will present the different functionalities in CuentosIE: (a) the search for tales; (b) the chat about emotions; (c) the addition of new tales. User interactions with CuentosIE are facilitated by the discourse manager, whose intricacies are elucidated in 'The Discourse Manager and Question Generator for Reading Comprehension Modules'. Functionality (a), corresponding to flow (2), is implemented on tales carefully selected, classified, and labeled by our team of psychologists based on emotions and psychological themes to ensure their effectiveness in emotion education (contribution (C4)). This functionality is comprehensively presented in 'The Information Retrieval (IR) Module: Selection, Classification and Labeling of Tales'. Functionality (b) falls under the purview of the Emotion Classifier module, extensively described in 'The Emotion Classifier Module'. Finally, functionality (c), pertaining to flow (4), enables users to contribute new tales to CuentosIE. These tales, upon evaluation and approval by our psychologists, are incorporated into the Information Retrieval module. This mechanism allows users to express their interests in novel tales and emotions, potentially captivating other users.

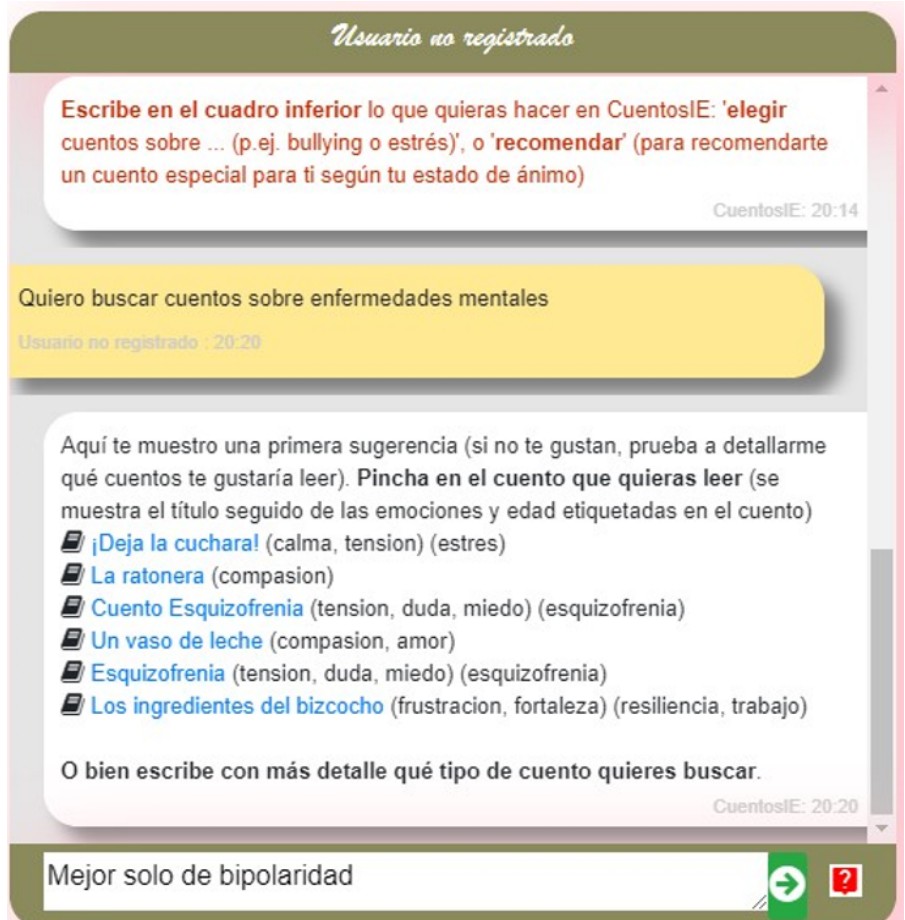

**Figure 2** An example of user's searches that can be refined successively.

### The discourse manager and question generator for reading comprehension modules

The user's interactions with CuentosIE are managed by various NLP modules that collaborate with each other. The Discourse Manager (DM), which is contribution (C3), handles user intention detection by running the appropriate NLP module, such as the Information Retrieval (IR) engine, the Question Generator for reading comprehension, or the Emotion Classifier engine (all of which are shown in the innermost box of Fig. 1). The DM consists of a classifier developed in Python that utilizes the textual input provided by the user and classifies it into one of the following five categories: the three CuentosIE functionalities (from (a) to (c)), the "exit of the application" class, and the "no_intention" class (which indicates that none of the previous classes were detected). This classifier was trained in the same way as the emotion classifier, which will be described in 'The Emotion Classifier Module'. Additionally, to improve CuentosIE's usability, a help icon (🔳 in Fig. 2)

is always available. This icon opens a contextual menu with various options (such as switching to a different functionality).

For example, when the user is in functionality (a), searching for tales, each user input is processed by the discourse manager (DM) to determine the user's intention. If the user expresses a desire to switch to functionality (b) (*e.g.*, "I'm tired of looking for tales, I would like to talk to you about emotions"), the DM will transfer control to the Emotion Classifier engine. Otherwise, if the "no_intention" class is detected, the user input is redirected to the appropriate NLP module. For example, in Fig. 2, the IR engine will receive "I want to search for tales on mental illnesses" as a search query. After the chatbot presents a list of the most relevant tales on this topic, the user posts a new search in order to refine it successively: "Better only on bipolarity".

When a user selects a tale to read (see (3) in Fig. 1), CuentosIE posts the text of the tale in the chatbot (🗩), along with its URL. Additionally, CuentosIE can read the tale aloud (🔊), which can be helpful for users with accessibility issues. Finally, the chatbot stores statistics on the tales, emotions, and psychological themes that CuentosIE users read. These statistics will be discussed in contribution (C5) in 'The Emotional Evolution Monitoring Module'.

After CuentosIE presents the tale content to the user, a conversation begins with a set of questions generated by the "Question Generator for reading comprehension" module in Fig. 1. Some of these questions are pre-designed and compiled by our psychologists, based on criteria related to their utility for improving emotion knowledge, improving the user's level of reading comprehension, and making the user reflect on the emotions that the tale deals with. Some of the questions are closed-ended (*e.g.*, "Do you think that this tale deals with 'frustration and strength' emotions?"), while others are open-ended to make the user think about the content of the tale (*e.g.*, "What are your feelings after reading the tale?"). In any question, new questions are posed to achieve active listening in the chatbot conversation. For example, after the closed-ended question, if the user's response is negative, the question "OK, then tell me which emotions you think that tale deals with" is proposed. For the open-ended question example, the CuentosIE emotion classifier analyzes the user's answer, questioning the user if their feelings are about the emotion classes detected by the classifier: "I guess that your feelings are related to 'sadness', am I right?". Other questions are automatically generated offline for each tale using the NLP tools mentioned previously. For example, the named entities in the tale are detected and questioned: "Who with his face downcast with regret, meets with his friend Marisa in a bar to have a coffee?".

During this conversation, active listening is also performed using large language models (LLMs) from modern chatbots such as ChatGPT or BARD. We specifically used the Falcon LLM (*Almazrouei et al., 2023*, https://falconllm.tii.ae/falcon.html). For example, for the conversation: "[CuentosIE] Who would you recommend this tale to? [User] To my children, my friends, my family, actually to all the people I know and even more so to the people who I consider interested, empathetic and with little sensitivity", several prompts are sent to Falcon: "[Prompt 1] Paraphrase the following sentence showing empathy: [user's response]"; or "[Prompt 2] How do you feel about this sentence: [user's response]". Falcon generates the following question, which is presented to the user: "I see empathy

towards your loved ones, acquaintances, and strangers alike, especially those who possess qualities of interest, empathy, and sensitivity, have I got that right?''. Falcon is also used to automatically answer questions related to the tale, in order to challenge the user to contrast their answer. For example, for the CuentosIE question ''What part of the tale did you like the most?'', Falcon would generate the following prompt: ''[Prompt] [CuentosIE_question] of this tale in quotes '[the content of the tale]'''. These LLMs are also used to leverage their great ability to summarize. For example, after the user has summarized the tale, Falcon can be used to generate a summary of the tale using the following prompt: ''[Prompt] Summarize this tale in quotes '[the content of the tale]'''. The summary generated by Falcon is then presented to the user along with their own summary, and the user is asked to reflect on both versions of the summary. This process helps to control the hallucination issue in modern chatbots by restricting their operation to the pieces of knowledge that have been previously chosen by psychologists: contribution (C2).

The user can interrupt this question loop at any time by expressing their intention in natural language (*e.g.*, ''I'd like to select a new tale''). This is detected and handled by the DM module. The user can also interrupt the loop by clicking the help icon (❓) in Fig. 2.

## The Information Retrieval module: selection, classification and labeling of tales

The IR engine has previously indexed the tales that have been curated by our psychologist authors (see (2) in Fig. 1 and contribution (C4)). The user's search can refer to the titles, content, emotions, or psychological themes of the tales. Additionally, the IR engine in CuentosIE uses NLP techniques such as a POS-tagger, partial parser, and semantic knowledge, as described in *Ferrández (2011)*. These techniques are combined with traditional IR techniques such as the Deviation from Randomness (DFR) measure (*Amati, Carpineto & Romano, 2004*).

Developed in C++ for optimal efficiency, this engine processes textual user queries and delivers sorted, relevant tales, accompanied by the emotions and psychological themes they elucidate. For example, in Fig. 2, the tale ''Los ingredientes del bizcocho (frustración, fortaleza) (resiliencia, trabajo)'' (translated as ''The Ingredients of the Cake'') has been manually tagged with the emotions ''frustration'' and ''strength'' and the psychological themes ''resilience'' and ''work''. We chose the taxonomy of 30 emotions proposed by *Díaz & Flores (2001)* to label the tales because we consider it to have enough depth to cover the most significant emotions. Additionally, *Díaz & Flores (2001)* proposes a set of terms associated with each emotion, which we will use in our emotion classifier. We selected a subset of the psychological themes exposed in the American Psychological Association (APA, https://www.apa.org/topics/) as representatives of current problems that could be of interest to people (*e.g.*, depression, resilience, stress, addiction, abortion, sex, adolescence or bullying). Since the main aim of CuentosIE is to guide the user in their knowledge about emotions, each emotion and psychological theme is accompanied by their definitions, related terms, and explicative videos curated and tagged by our psychologist authors.

Additionally, users can add new tales to CuentosIE (functionality (c) in Fig. 1). Each added tale is reviewed by our psychologists to ensure the quality and usefulness of the tales

indexed in the chatbot: *contribution (C4)*. These tales are also manually tagged according to our taxonomy of emotions and psychological themes. This will create a corpus of tales tagged with emotion and psychological knowledge, which will be made available to the scientific community in XML format (🗄) in (5) in Fig. 1. This corpus will be especially valuable for machine and deep learning techniques, which require large and tagged datasets.

## The Emotion Classifier module

Regarding the (b) functionality in CuentosIE (the chat about emotions), it is implemented in the Emotion Classifier module and primarily aims to achieve emotion education objectives. In this functionality, CuentosIE poses a starting question to the user to encourage them to talk about their current emotional state (*e.g.*, ''Hello, how are you today? Please, let's talk about whatever you want. The more we chat, the better I can recommend a tale that's right for you''). After each user post, the Emotion Classifier engine returns the most salient emotion in the post and answers the user, asking for confirmation about this emotion classification. An example extracted from the experiments in the following section starts with the user's post: ''Tonight I had insomnia''. CuentosIE classifies the user's post as dealing with the ''tension'' emotion, so it answers: ''Do you know that the emotion 'tension' is defined as: 'Feeling of restlessness, discomfort'? Do you think that your current emotional state is identified with this emotion?'' followed by the quote: ''An interesting quote to reflect on: 'Insomnia is an extra time that life gives us, to keep thinking about what hurts so much.'''. These quotes have been previously curated by our psychologists and indexed separately from the tales by the IR engine. After that, the conversation continues until the user decides to end it (which is detected by the DM module) in natural language (*e.g.*, ''We've talked enough, please, recommend me a tale'') or by clicking the help icon 🔴 in Fig. 2. Then, CuentosIE compiles the emotions that have been detected in the conversation and recommends tales that deal with these emotions to the user.

Our emotion classifier has been evaluated using various machine learning models implemented in Python, including the scikit-learn and TensorFlow libraries: Decision Tree, Random Forest, Naïve Bayes, Support-Vector Machine (SVM) and the Transformer model ''tf_roberta_for_sequence_classification'' (https://huggingface.co/PlanTL-GOB-ES; with 124,644,866 parameters). Our evaluation revealed that the Transformer model outperformed the remaining machine learning models. The classifier distinguishes between the 30 emotions proposed by *Díaz & Flores (2001)*. It was trained on: (1) the set of terms linked to each emotion in *Díaz & Flores (2001)*; (2) the Wikipedia pages of each emotion; (3) definitions of each emotion extracted from the web; (4) synonyms of each emotion extracted from WordNet and WordReference. The evaluation on these four sets of tagged corpora achieved an accuracy of 84.53%. The classification task is challenging due to the high number of classes (30 emotions) and the fact that some emotion descriptions explain the relationships between similar or opposite emotions. This makes the task more difficult for the classifier. The following passages from Wikipedia illustrate the ambiguity between the vocabulary of close or opposite emotions:

- Sadness is an emotional pain associated with, or characterized by, feelings of disadvantage, loss, despair, grief, helplessness, disappointment and sorrow. An individual

experiencing sadness may become quiet or lethargic, and withdraw themselves from others. An example of severe sadness is depression, a mood which can be brought on by major depressive disorder or persistent depressive disorder. Crying can be an indication of sadness (*Wikipedia, 2024a*).

- In psychology, stress is a feeling of emotional strain and pressure.[1] Stress is a type of psychological pain. Small amounts of stress may be beneficial, as it can improve athletic performance, motivation and reaction to the environment. Excessive amounts of stress, however, can increase the risk of strokes, heart attacks, ulcers, and mental illnesses such as depression[2] and also aggravation of a pre-existing condition. Stress can be external and related to the environment,[3] but may also be caused by internal perceptions that cause an individual to experience anxiety or other negative emotions surrounding a situation, such as pressure, discomfort, *etc.*, which they then deem stressful (*Wikipedia, 2024b*).

- Comfort (or being comfortable) is a sense of physical or psychological ease, often characterized as a lack of hardship. Persons who are lacking in comfort are uncomfortable, or experiencing discomfort …implies that the subject is in a state of pain, suffering or affliction (*Wikipedia, 2024c*).

Table 1 shows some examples of user interactions in this functionality, which were obtained from the CuentosIE experimentation to be reported in 'Experimentation'. To ensure transparency, user posts are first presented in Spanish as originally posed, followed by their English translation in parentheses (in subsequent tables and examples, user posts will be presented only in English for clarity). It is worth noting the high number of negations present in the sentences (*e.g.*, numbers 2, 3, 4, 7, and 8). Additionally, some examples deal with contradictory emotions, such as snippet 8, which deals with negative initial emotions and positive present emotions. Furthermore, there are some spelling errors in the sentences, which makes the classification process more difficult. For example, in snippet 6, the adverb "yes" is misspelled, as "si" (which means "if" in English) should be "sí" in Spanish. Finally, we highlight the need to address some complex linguistic issues, such as anaphora, as these examples are embedded in a conversation with several interactions (*e.g.*, snippet 1, with the anaphoric reference "that" to previous interactions in its conversation).

### The emotional evolution monitoring module

In this final subsection, we describe the Emotional Evolution Monitoring module that deals with the CuentosIE's ability to store user conversations and interactions: contribution (C5), in flows (3) and (5) in Fig. 1. These conversations are stored in XML files, along with the timestamp of each interaction. This allows us to track the user's progress over time, which can be useful for psychologists and teachers.

As an example, here is an excerpt from an XML conversation during the questions that analyze the tale: <interaction><date>25/05/2023 14:41:00</date><user>atg9</user><CuentosIE>Tell me if you would have done the same or something similar</CuentosIE><answer>Yes, sometimes in some circumstances it is difficult for me to ignore harmful comments that do not add up, but in many others I ignore them and I

**Table 1 Examples of the users' conversations about "emotions".**

| | |
|---|---|
| 1 | Simplemente eso aunque he dormido poco (Just that although I have slept little) |
| 2 | No estoy teniendo muchas emociones de tristeza pero estoy teniendo poca energía últimamente (I'm not having a lot of sad emotions but I'm having low energy lately) |
| 3 | Yo estoy cansado porque ayer no dormí bien (I am tired because yesterday I did not sleep well) |
| 4 | No no hay algo que me preocupe, de momento el día me está yendo bien (No no there is nothing that worries me, for the moment the day is going well for me) |
| 5 | Pasar tiempo con mis amigos (Spend time with my friends) |
| 6 | Si, y si me gustan (Yes, and if I like them) |
| 7 | La verdad es que no, no tengo emociones de humillación, la verdad no se porque no siento que nadie me juzgue (The truth is that no, I do not have emotions of humiliation, the truth is I do not know because I do not feel that anyone is judging me) |
| 8 | Pues hoy me he levantado mejor que muchos días y de momento nada está siendo tan duro. Si fuera así siempre estaría mucho mejor (Well today I woke up better than many days and at the moment nothing is being so hard. If it were so, it would always be much better) |

**Table 2 Examples of insightful excerpts of users' conversations.**

| | |
|---|---|
| 1 | I'm tired of living |
| 2 | Every day it is harder to continue |
| 3 | There are times I would like to end it all |
| 4 | I'm fine, the only thing that worries me is the grades, I know that my mother is happy and I'm happy about it, since my mother is the most important thing I have in this life, if she dies or something happens to her, I wouldn't be able to live with it |
| 5 | I am calm, and happy, and I am satisfied with the life I have, but deep down I am somewhat shy |
| 6 | The worst thing is that the last conversation with my father is that I yell at him, and he dies |

know that they do not add up to me and that they only want to sink and see how you do not move forward</answer></interaction>.

Table 2 shows other examples of real user conversations that demonstrate their engagement with the chatbot and how they feel comfortable sharing their emotions without fear of being judged, laughed at, or hurt. These XML files can be used to detect sensitive situations such as depression (*Havigerová et al., 2019*), suicide (*Boggs & Kafka, 2022*), and bullying (*Bayari & Bensefia, 2021*).

Another important piece of information that CuentosIE collects is statistics on the tales, emotions, and psychological themes selected by users (shown in ⌂ in (5) of Fig. 1). These statistics can be a valuable indicator of users' interests in the field of emotions, as will be shown in the experimentation section.

## EXPERIMENTATION

In this section, we analyze the experiments conducted to test the benefits of CuentosIE: contribution (C6). The hypothesis behind these experiments is to answer the question posed in the title of this article: "Can a chatbot about 'tales with a message' help to teach Emotional Intelligence?" This hypothesis will be evaluated by specifically evaluating the five remaining contributions described in the article.

## Materials & methods

In the experiments, we tested the applicability of CuentosIE with different ranges of users by dividing them into three groups with different academic and age levels: (i) secondary school students (under 18), (ii) university students (18–23), and (iii) post-graduate students (over 23). None of the users received any monetary compensation. The studies involving human participants were reviewed and approved by the Ethics Committee of the University of Alicante (Exp. UA-2020-05-12). Written informed consent to participate in this study was provided by the participants' legal guardian/next of kin. Group (i) consisted of students aged 13–16 from a private compulsory secondary education center in Malaga, Spain. The experiment was conducted at the school as an educational activity during their tutorial subjects. There were 25 students per classroom, and the class tutor, computer science teacher, and one of our psychologist authors were present. The students first received a brief description of the experiment's purpose, CuentosIE's goals and operating mode, and general concepts about emotional intelligence. Then, they conversed with the chatbot individually using an iPad. The testing was carried out through four face-to-face sessions, each lasting one hour and held every 15 days at the school. Groups (ii) and (iii) were recruited in a similar way, as an activity during the subjects lectured by our psychologists. They consisted of university students of social work in the first year, education in the first, second, and third years, and post-graduate students of the Master in Psychopedagogy at the University of Alicante. Since the users were advised that all interactions would be stored for further analysis, no user was forced to register in CuentosIE. As a result, only 360 users registered, as shown in Fig. 3, with a distribution of males, females, and ages. The following experiments will be segmented by these parameters to draw significant conclusions, as per *contribution (C5)*. All the users interacted with the chatbot while the psychologist answered any questions they had about using the application and encouraged them to freely test the different functionalities, especially those of "tale search and analysis" and "talking about emotions".

## RESULTS & DISCUSSION

After users worked with CuentosIE, they were encouraged (not forced) to complete a survey form that includes quantitative and qualitative feedback (*Creswell, 2022*). The quantitative results are summarized in Table 3. The survey questions were designed to measure both the conversational agent's efficiency, effectiveness, and functionality, as well as users' satisfaction with their improvement in emotional intelligence. Users ranked the following issues from 0 to 10 (as shown in Table 3): their overall feeling about the tool, their opinion on the search and suggestion process for tales, their chat process about emotions, and their self-evaluation of their emotional intelligence improvements after using CuentosIE. The 727 surveys reported in this table are more than the 360 registered users because some users who did not sign up for CuentosIE also completed the survey. After analyzing Table 3, we can conclude that users achieved a remarkable overall satisfaction (score of 7.82), with no significant differences between male and female users, but slightly lower scores for users under the age of 18. These results hold for the detailed scores of the tool's options, although

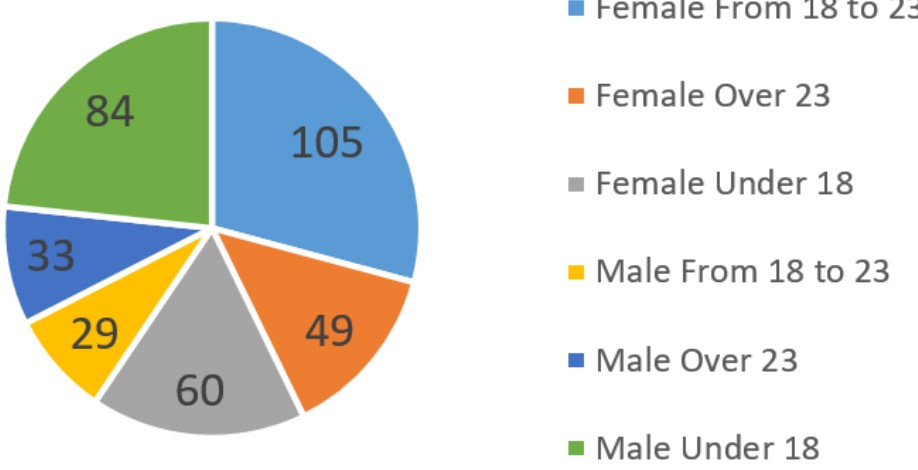

- Female From 18 to 23
- Female Over 23
- Female Under 18
- Male From 18 to 23
- Male Over 23
- Male Under 18

**Figure 3** **Male and female distribution in the 360 registered users in CuentosIE.**

**Table 3** **The 727 survey results (from 0 to 10) filled out by users after running CuentosIE.**

|  | Overall | Opinion on the search process | Opinion on the chat process | Self evaluation on Em.Intellig. improvement | #Surveys |
|---|---|---|---|---|---|
| Male | 7.93 | 7.86 | 7.53 | 6.78 | 341 |
| From 18 to 23 | 8.29 | 7.53 | 7.71 | 7.35 | 17 |
| Over 23 | 8.33 | 8.82 | 8.55 | 7.25 | 12 |
| Under 18 | 7.89 | 7.85 | 7.48 | 6.73 | 312 |
| Female | 7.73 | 7.93 | 7.52 | 6.56 | 386 |
| From 18 to 23 | 8.25 | 8.11 | 8.12 | 7.12 | 75 |
| Over 23 | 8.67 | 8.94 | 9.06 | 7.78 | 18 |
| Under 18 | 7.54 | 7.82 | 7.27 | 6.34 | 293 |
| Total | 7.82 | 7.90 | 7.52 | 6.66 | 727 |

the "talk about emotions" option (which runs the emotion classifier) received a slightly lower score (7.52 *vs.* 7.90 for the tale search option). Finally, the score given by users on how they feel the chatbot helps them improve their emotional intelligence is slightly lower than the overall score (6.66 *vs.* 7.82).

Regarding the qualitative feedback from users, the overall impressions from the experiments were positive, as participants freely chose to continue talking to CuentosIE after each session. These impressions were corroborated by the quantitative results and the following textual comments from the surveys: "I find it very useful and reading the tale you realize many things that you feel"; "It seems to me to be a way to get a person to connect with their emotions, and to know their interior and how they feel through a tale"; "I find it very useful and I would recommend it to my friends". Negative feedback mainly focused on the lack of a greater variety of tales and the instances in which the emotion classifier

failed: "It is a useful and quite intelligent application, it asks interesting curious questions but it needs to have a larger repertoire of books"; "I would add drawings to associate the emotion with an image and so the little ones in sixth or fifth grade could do it"; "When you are talking to the machine sometimes it gives incoherent emotions".

Based on these results, we believe that contributions (C2) and (C3) have been met, as users have not complained about the "hallucination" issues present in modern chatbots, the different NLP modules interact correctly, and the survey scores have been satisfactory. Similarly, contribution (C4), which deals with the selection, classification, and labeling of tales according to emotions and psychological themes as the core of the chatbot for teaching emotions, can be considered complete. However, this contribution should be improved in the future, as only 50 tales were indexed in CuentosIE during the experiments, which resulted in some emotions and psychological themes not being addressed by any tales. This was also corroborated by some user comments: "Perhaps the link between the tales and the emotions and psychological themes discussed should be polished a little more. Regarding psychological issues, they could also be expanded further".

Regarding the contribution (C5), as an example of the indicators and analysis that can be extracted from the interactions stored in CuentosIE, Figs. 4 and 5 present the percentage of each emotion (the extensive spectrum of 30 emotions that CuentosIE effectively encompasses) selected by users, segmented by gender (female in Fig. 4, and male in Fig. 5), and age distributions (From 18 to 23; over 23; under 18). These percentages were calculated based on the 5,624 emotions chosen by users to ask CuentosIE to show them tales that deal with those emotions. We grouped positive emotions as "joy, desire, certainty, strength, enthusiasm, calm, pleasure, love, courage, fun, liking, compassion, and satisfaction", and negative emotions as "tension, phobia, boredom, humiliation, discomfort, sadness, apathy, doubt, pain, frustration, hatred, exhaustion, emotional dependency, attachment, fear, arrogance, and anger". After analyzing the grouped emotions, we observed that "positive" emotions are more in demand than "negative" emotions: 56% *vs.* 44%. However, the difference shows comparable levels of interest between them. The results obtained by females and males present interesting conclusions, where the highest selected emotions are "joy and calm" for both, but there are differences in the "tension, exhaustion, and doubt" emotions (higher for females). Concerning the analysis of the age groups, we can highlight substantial differences. For example, the highest selected emotions for users under 18 are "joy and calm", whereas for the remaining groups, "tension and doubt" are the most in demand. Thanks to this data, which is linked to each registered user, CuentosIE can obtain indicators about users' interests in specific emotions, as well as analyze the timeline evolution of users' emotion interests at different points in time.

Based on these experimental results, we believe that the initial hypothesis is supported: interacting with the chatbot has been beneficial to users, and tales have been an effective way to teach emotional intelligence. As a future line of research, these results should be corroborated by measuring the level of development of emotional intelligence in a group of users using a psychometric tool (*e.g.,* the one proposed by *Mayer, Salovey & Caruso, 2016*). After measuring their level of development in emotional intelligence with the psychometric tool, we will carry out a one-and-a-half-month intervention program using the CuentosIE

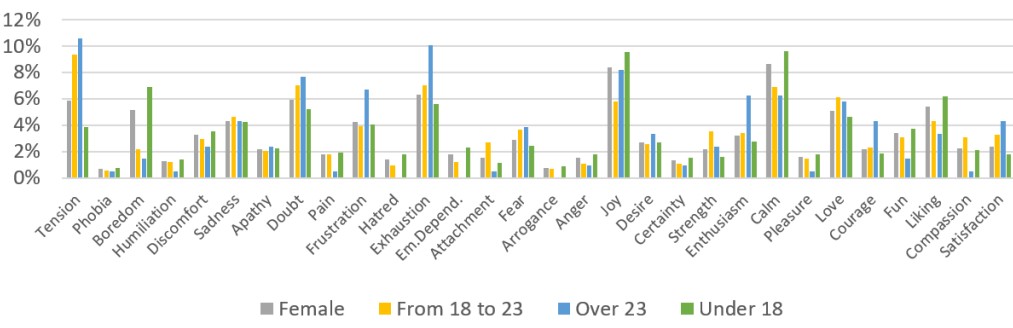

**Figure 4** Emotions chosen to select tales by female users, for different age ranges.

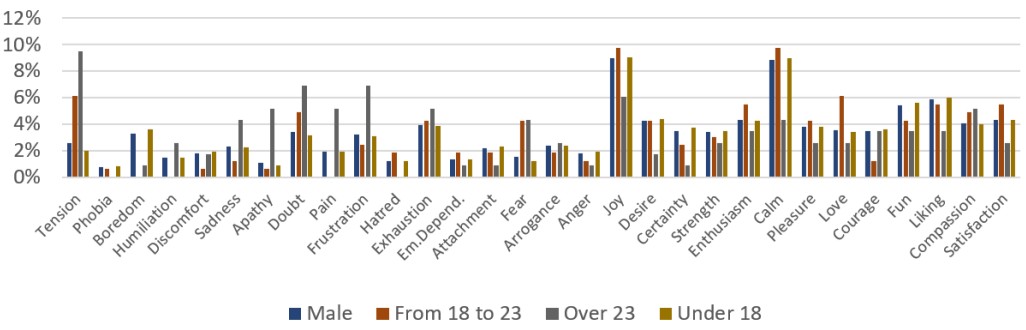

**Figure 5** Emotions chosen to select tales by male users, for different age ranges.

tool. After this period, we will again measure the level of development of emotional intelligence of the participants to verify whether CuentosIE enhances the improvement of the metacognitive skills associated with emotional intelligence.

## CONCLUSIONS

In this article, we present the chatbot CuentosIE. It facilitates wide availability and anonymity through its web-based interaction and specializes in "tales with a message", which have been shown to be highly effective through their moral or associated metaphors (contribution (C1), detailed in the Introduction section). CuentosIE includes an information retrieval system, a reading comprehension question generator, and an emotion classifier system, all of which are designed to develop users' emotional intelligence. It avoids the hallucination issue present in modern chatbots such as ChatGPT and BARD, which could be critical in education fields, by restricting their application to specific non-critical tasks and pieces of knowledge (contributions (C2) and (C3)). Additionally, our team of psychologists has selected, classified, and labeled these tales according to emotions and psychological themes to ensure their utility in teaching emotions (contribution (C4)). All user interactions with the chatbot are stored in XML files, which allows for further processing to detect sensitive situations (contribution (C5)). The CuentosIE experimentation was conducted on users from three different academic and age levels to

cover a variety of user types, with encouraging results. These groups allow us to segment the data by these parameters to draw significant conclusions (contribution (C6)).

Based on the preliminary results of this pilot study, in the future, we plan to evaluate the personality profile of the participants to determine whether there are correlations between their personality profiles and the types of emotions they are most interested in. Additionally, given that there appear to be differences in interest in emotions according to gender, we will explore this further in our new data collection, which will include a larger sample size.

### Funding

The research work conducted is part of the R&D projects ''CORTEX: Conscious Text Generation'' (PID2021-123956OB-I00), funded by MCIN/ AEI/10.13039/501100011033/ and the project ''NL4DISMIS: Natural Language Technologies for dealing with dis- and misinformation with grant reference (CIPROM/2021/021)'' funded by the Generalitat Valenciana. Furthermore, it has been funded by the BALLADEER project (PROMETEO/2021/088) from the Consellería Valenciana and by the AETHER-UA (PID2020-112540RB-C43) project from the Spanish Ministry of Science and Innovation. The funders had no role in study design, data collection and analysis, decision to publish, or preparation of the manuscript.

### Grant Disclosures

The following grant information was disclosed by the authors:
R&D projects ''CORTEX: Conscious Text Generation'': PID2021-123956OB-I00.
MCIN/ AEI/10.13039/501100011033/.
Natural Language Technologies for dealing with dis- and misinformation (CIPROM/2021/021): NL4DISMIS.
Generalitat Valenciana. BALLADEER project: PROMETEO/2021/088.
Consellería Valenciana AETHER-UA: PID2020-112540RB-C43 project from the Spanish Ministry of Science and Innovation.

### Competing Interests

The authors declare there are no competing interests.

### Author Contributions

- Antonio Ferrández conceived and designed the experiments, performed the experiments, analyzed the data, performed the computation work, prepared figures and/or tables, authored or reviewed drafts of the article, and approved the final draft.
- Rocío Lavigne-Cerván conceived and designed the experiments, performed the experiments, authored or reviewed drafts of the article, and approved the final draft.
- Jesús Peral analyzed the data, authored or reviewed drafts of the article, and approved the final draft.

- Ignasi Navarro-Soria conceived and designed the experiments, performed the experiments, authored or reviewed drafts of the article, and approved the final draft.
- Ángel Lloret analyzed the data, authored or reviewed drafts of the article, and approved the final draft.
- David Gil analyzed the data, authored or reviewed drafts of the article, and approved the final draft.
- Carmen Rocamora conceived and designed the experiments, performed the experiments, authored or reviewed drafts of the article, and approved the final draft.

### Ethics

The following information was supplied relating to ethical approvals (*i.e.*, approving body and any reference numbers):

The studies involving human participants were reviewed and approved by the Ethics Committee of the University of Alicante.

### Data Availability

The raw data is available in the Supplemental File.

The code is also available in the Supplemental Files and at https://oldgplsi.gplsi.es/cuentosIE/index.php and https://oldgplsi.gplsi.es/cuentosIE/chatbot_CuentosIE.php.

### Supplemental Information

Supplemental information for this article can be found online at http://dx.doi.org/10.7717/peerj-cs.1866#supplemental-information.

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
