# Peer review of "CuentosIE: can a chatbot about “tales with a message” help to teach emotional intelligence?"

_PeerJ Computer Science, doi:10.7717/peerj-cs.1866_

## Round 0.1 · original submission · Major Revisions

While both reviewers' feedback are detailed, reviewer 2's feedback is slightly more complex. In the "Validity of the findings", Reviewer 2 raised the issue that it is written as a psychology paper and not enough details are provided to make it a computer science paper.

If the authors give a detailed explanation of its system architecture when revising it, this will make it more in scope for computer science, i.e. detail the architecture and the code you are using and place less emphasis on the psychology aspect. This solution is not immediately clear from Reviewer 2 but addressing it is necessary if the submission is to be considered suitable for PeerJ Computer Science.

Please address the issues raised by the reviewers and provide a revised manuscript.

Reviewer 1 ·

Basic reporting

The paper first provides the motivations, for why they build the chatbot CuentosIE (contains from C1 to C6 contributions). Then, the paper provides a deeper background on dealing with emotions and how chatbots are used for emotion processing. Furthermore, the paper illustrates the design and implementation details of this proposed chatbot CuentosIE. The Experiment section shows how they set up the experiment and how the chatbot CuentosIE meets the contributions listed in C1 - C6. The paper offers a clear motivation and definitions of all terms and theorems.

Experimental design

The overall design of experiments is reasonable. But it still needs some clarifications: 1) Why did the emotion classifier pick the Transformer model "tf_roberta_for_sequence_classification"? Did you cherry-pick the Transformer model here? 2) For Figures 4 and 5, are the 5,624 emotions enough to generate an accurate experimental result?

Validity of the findings

The experimental results are well-supported.

Additional comments

Please add more references to some parts in the paper, such as:
(line 196) "every day more companies include a chatbot on their websites or social networks to offer products and services to customers..." Please list one or two examples here.
(line 197) "Although most chatbots have been developed for commercial purposes, these machines are also very useful in other areas..." (such as the existing work "DroidPerf: Profiling Memory Objects on Android Devices" [MobiCom'23] develops a communicable bot to evaluate memory issues on smartphone terminal);
(line 280) "NLTK and the spaCy library"
(Figure 4 and 5) Please add the reference of the used in the caption.

Reviewer 2 ·

Basic reporting

The paper used clean and understandable English to present it's idea. It was nicely written.

Experimental design

All of the experiments seems to come from a psychological experimental perspective. The experiments are from human responses rather than on actual datasets. Also there should be a much more detailed explanation of the system structure.

Validity of the findings

The paper presents an E2E chatbot system using traditional NLP techniques (NLTK, etc) with newer ones (Falcon LLM) that chats with people, especially those with mental issues, about their feelings. However it lacks detailed explanation of it's system architecture. It feels more like a psychology paper not an computer science paper.

---

## Round 0.2 · accepted · Accept

The authors have sincerely addressed the issues raised by the reviewers. I now recommend accepting the manuscript for publication.

Reviewer 1 ·

Basic reporting

The paper first provides the motivations, for why they build the chatbot CuentosIE (contains from C1 to C6 contributions). Then, the paper provides a deeper background on dealing with emotions and how chatbots are used for emotion processing. Furthermore, the paper illustrates the design and implementation details of this proposed chatbot CuentosIE. The Experiment section shows how they set up the experiment and how the chatbot CuentosIE meets the contributions listed in C1 - C6. The paper offers a clear motivation and definitions of all terms and theorems.

Experimental design

The overall design of experiments is reasonable.

Validity of the findings

The experimental results are well-supported.

Additional comments

The revised version addressed all my concerns/questions in the first round of review.